# Comparative Composition Structure and Selected Techno-Functional Elucidation of Flaxseed Protein Fractions

**DOI:** 10.3390/foods11131820

**Published:** 2022-06-21

**Authors:** Xiaopeng Qin, Linbo Li, Xiao Yu, Qianchun Deng, Qisen Xiang, Yingying Zhu

**Affiliations:** 1College of Food and Bioengineering, Zhengzhou University of Light Industry, Zhengzhou 450002, China; 15083059355@163.com (X.Q.); 15517587171@163.com (L.L.); xiangqisen2006@163.com (Q.X.); zhuying881020@163.com (Y.Z.); 2Oil Crops Research Institute, Chinese Academy of Agricultural Sciences, Wuhan 430062, China

**Keywords:** flaxseed protein fractions, component structure, techno-functionality, phenolic acids, interfacial behavior

## Abstract

This study aimed to comparatively elucidate the composition structure and techno-functionality of flaxseed protein isolate (FPI), globulin (FG), and albumin (FA) fractions. The results showed that FA possessed smaller particle dimensions and superior protein solubility compared to that of FG (*p* < 0.05) due to the lower molecular weight and hydrophobicity. FA and FG manifested lamellar structure and nearly spherical morphology, respectively, whereas FPI exhibited small lamellar strip structure packed by the blurring spheres. The Far-UV CD, FTIR spectrum, and intrinsic fluorescence confirmed more flexible conformation of FA than that of FG, followed by FPI. The preferential retention of free phenolic acids was observed for FA, leading to excellent antioxidant activities compared with that of FG in FPI (*p* < 0.05). FA contributed to the foaming properties of FPI, relying on the earlier interfacial adsorption and higher viscoelastic properties. FA displayed favorable emulsifying capacity but inferior stability due to the limited interfacial adsorption and deformation, as well as loose/porous interface. By comparison, an interlayer anchoring but no direct interface coating was observed for lipid droplets constructed by FG, thereby leading to preferable emulsion stability. However, FPI produced lipid droplets with dense interface owing to the effective migration of FA and FG from bulk phase, concomitant with the easy flocculation and coalescence. Thus, the techno-functionality of flaxseed protein could be tailed by modulating the retention of albumin fraction and specific phenolic acids.

## 1. Introduction

Considering milk allergies, lactose intolerance, vegetarians, and vegans, the transition from diets based on animal proteins towards diets primarily based on plant proteins has been occurring. Thus, the plant-based proteins with high functionality have been screened with respect to stabilize the multiphase food systems. Besides the bioactive α-linolenic acid, gum polysaccharides, lignans, and phenolic acids, flaxseed also contains appreciable amounts of protein with high nutritional quality [1]. Importantly, flaxseed protein and its hydrolysates possess desirable anti-hypertensive, anti-bacterial, and anti-diabetic activities, and also mitigated against ethanol or lead induced hepatotoxicity [2,3,4]. The techno-functionality of flaxseed protein, including water/oil-holding capacity, foaming, and emulsifying properties, had also been explored in previous studies [5,6]. Indeed, the naturally occurring gum polysaccharides could largely affect the hydrodynamic, foaming, and emulsifying properties of flaxseed protein obtained from whole or defatted flaxseed meal due to the noncovalent interaction between them [7,8]. Thus, the actual responsiveness between the molecular structure, spatial conformation, and techno-functionality of flaxseed protein and its fractions was still blurred [9]. This was particularly unfavorable for the optimal application of flaxseed-derived proteins in food-grade colloidal systems with desirable appearance or texture.

Whole flaxseed has an average protein content of 23%, containing approximately 70~85% of salt-soluble globulin and 15~30% of water-soluble albumin fractions, relying on the extraction substrates and methods [10,11,12]. The multiple polypeptides with higher molecular weight (Mw), hydrophobic and branched-chain amino acids were observed for globulin, whereas a single polypeptide with abundant negatively charged and sulfur-containing amino acids was prominent in albumin fraction [2]. As previously reported, flaxseed globulin and albumin fractions exhibited discrepant foam/lipid droplet formation capacity and stability [6,13]. Theoretically, protein fractions could spontaneously form the air–water or oil–water interfaces following the interfacial adsorption, structural reorganization, and sequential membrane expansion behavior against foam/lipid droplet instability [14]. Unfortunately, it was still undefined as to how the heterogeneous composition and structure between globulin and albumin fractions in flaxseed with stripping of gum polysaccharides affected their stabilizing and destabilizing properties of foam and lipid droplets based on the interfacial behaviors.

Increasing evidence had revealed that the phenolic compounds could naturally exist in protein fractions extracted from flaxseed due to the ionic bond, hydrogen bond, or hydrophobic interactions with phenolic hydroxyl and carboxyl groups [15]. The retention for phenolic compounds, including lignans and phenolic acids, could further affect the water-holding capacity or gel properties of flaxseed protein prepared by micellization and/or isoelectric precipitation [16]. Thus, the exogenous addition flaxseed polyphenols via simple complex or covalent binding had been conducted to positively regulate the antioxidant and emulsifying properties of flaxseed protein [17,18]. In particular, the free phenolic acids might be coexisted with globulin and albumin in protein bodies or embedded into membrane proteins of oil bodies against the in situ lipid oxidation of α-linolenic acid in flaxseed [19]. Actually, due to the distinct composition structure and spatial conformation between globulin and albumin, how the specific accumulation of phenolic compounds and ascending antioxidant potential occur was still undefined following the sequential extraction of flaxseed protein fractions [20]. Based on the above, the aim of this work was to conduct a comparative techno-functional elucidation, including the antioxidant, foaming, and emulsifying properties of flaxseed protein and its fraction globulin and albumin, focusing on their component structure, retention of phenolic acids, interfacial, and rheological behavior. This discrepancy in specific techno-functionality for flaxseed protein fractions could achieve the tailed application in foam- and emulsion-type foods, such as whipped cream, ice cream, mayonnaise, dressing, margarine, etc.

## 2. Materials and Methods

### 2.1. Chemicals and Materials

Dehulled flaxseed (variety: longya 13 #) was provided by Gansu academy of agricultural sciences (Lanzhou, China). Flaxseed oil was purchased from Hongjingyuan oil Co., Ltd. (Xilingol, China). Folin–ciocalteu reagent, rutin, and amino acid standards were purchased from were purchased from Beijing Solarbio Sciences and Technology Co., Ltd. (Beijing, China). 2,4,6-Tris(2-pyridyl)-striazine (TPTZ, 99%) and 2,2′-diphenyl-1-picrylhydrazyl radical (DPPH, 95%) were obtained from Shanghai Beyotime Biotechnology Co., Ltd. (Shanghai, China). Vanillin, coumarin, p-coumaric, gallic, protocatechuic, vanillic, caffeic, syringic, sinapic, ferulic, sallcylic, and cinnamic acids were provided by Shanghai Yuanye Biological Technology Co., Ltd. (Shanghai, China). Micro BCA Protein Assay Kit was obtained from Beyotime Biotechnology Co., Ltd. (Shanghai, China). Other analytical grade reagents were purchased from Sinopharm Chemical Reagent Co., Ltd. (Beijing, China).

### 2.2. Extraction and Isolation of Protein Fractions from Dehulled Flaxseed

The dehulled flaxseed was ground into fine powder using a coffee grinder, defatted twice using hexane at a ratio of 1:4 (*w*/*v*) for 2 h, and filtered using a filter paper disk. After being air-dried in a fume hood overnight, part of the defatted powder was dispersed in deionized water at a ratio of 1:15 (*w*/*v*), adjusted to pH 8.5 using 0.5 M NaOH solution, and stirred at 500 rpm for 2 h at 25 °C. Then, the supernatant was obtained following centrifugation at 8000× *g* for 20 min, adjusted to pH value of 3.8 with 0.5 M HCl to precipitate protein [21]. After centrifugation at 8000× *g* for 20 min again, the protein precipitation was dispersed in deionized water and adjusted to pH value of 6.8. In order to remove the non-bound salt ions, the above protein dispersion was subjected to dialysis using a dialysis bag with molecular cut off weight of 8–10 kDa against deionized water for 24 h at 4 °C, and then freeze-dried to obtain flaxseed protein isolate (FPI). The remaining part of defatted powder was dispersed in deionized water at 1:15 ratio (*w*/*v*) and stirred at 500 rpm for 2 h at 25 °C. After centrifugation at 8000× *g* for 20 min, collect the supernatant and repeat this step until the supernatant becomes transparent. The FA fraction was obtained after freeze-drying. The resulting precipitation was mixed, with 0.5 M NaCl at 1:10 ratio, stirred for 2 h at 25 °C and centrifuged at 8000× *g* for 20 min. The supernatant was subjected to dialysis using a dialysis bag with molecular cut off weight of 8–10 kDa, and then freeze-dried to obtain flaxseed globulin (FG) fraction [6].

### 2.3. Analysis of Proximate Composition of Flaxseed Protein Fractions

The contents of crude protein in flaxseed protein fractions were determined by automatic kjeldahl analyzer and calculated with a conversion factor of 6.25 [5]. For the amino acid composition, the protein samples were hydrolyzed with 6 M HCl for 24 h at 110 °C in a sealed tube and analyzed using a Biochrom 30 automatic amino acid analyzer. The individual amino acid was identified, and the results were expressed as the percentage of moisture content. The levels of total lipids, ash, and moisture were determined according to the AOAC official methods 920.85, 923.03, and 930.15, respectively. The contents of total sugars were obtained by the subtraction of crude protein, total lipids, ash, and moisture in individual protein samples.

### 2.4. Analysis of Physicochemical Properties of Flaxseed Protein Fractions

The mean particle size and zeta potential of flaxseed protein fractions in 50 mM phosphate buffer solution (PBS, pH 6.8) were evaluated using a ZetaSizer Nano-ZS 90 (Malvern Instruments Ltd., Worcestershire, UK). For analysis of protein solubility, the protein samples were dispersed in 50 mM PBS (pH 6.8) at a concentration of 0.1% (*w*/*v*), continuously stirred for 2 h at 25 °C, and centrifugated at 6000× *g* for 20 min. The protein content in supernatant was determined by the MicroBCA assay using bovine serum albumin as the standard and dividing the total protein content in the supernatant by the protein content in the initial sample, which was used to calculate the percentage of the soluble protein relative to the total protein content [22]. For analysis of surface hydrophobicity, a series of dilutions was achieved by mixing flaxseed protein dispersion (0.1%, *w*/*v*) into 50 mM PBS (pH 6.8) ranging from 0.002% to 0.01% (*w*/*v*). Then, 20 μL of the 8-anilino-1-naphthalene sulphonic acid (ANS) solution (8.0 mM) was added into 4 mL of each dilution and mixed thoroughly using a vortex mixer. After reacting for 10 min in the dark at 25 °C, the fluorescence intensity (FI) was determined using a F-7000 fluorescence spectrophotometer (Hitachi, Tokyo, Japan) with excitation and emission wavelength at 390 nm and 480 nm, respectively. The surface hydrophobicity (H_0_) value was expressed by the initial slope of FI against sample concentration plot [23].

### 2.5. The Structural Properties of Flaxseed Protein Fractions

#### 2.5.1. Scanning Electron Microscope 

The surface morphology of flaxseed protein fractions was observed using high-resolution field emission scanning electron microscopy (FE-SEM) Regulus 8100 (Hitachi, Tokyo, Japan). Briefly, the power samples were attached to a sample stub with double-sided sticky tape, sputter coated with gold using a polaron sputter coater, and visualized at an accelerated voltage of 3.0 kV with magnification 10,000×.

#### 2.5.2. Sodium Dodecyl Sulphate-Polyacrylamide Gel Electrophoresis (SDS-PAGE)

The aqueous dispersions of flaxseed protein fractions with equal protein amounts (1.0 mg/mL) were mixed with a reducing (×4) loading buffer, heated at 95 °C for 5 min, and separated by 10% sodium dodecyl sulphate polyacrylamide gel electrophoresis (SDS-PAGE). The gels were stained with 0.1% of coomassie brilliant blue solution under gentle shaking, washed with destaining solution, and imaged on a ChemiDoc XRS+ System (Bio-Rad, Hercules, CA, USA).

#### 2.5.3. Intrinsic Fluorescence Spectra

The flaxseed protein fractions were diluted in deionized water (0.1 mg/mL) and analyzed for conformational characteristics by intrinsic fluorescence spectrum using a RF-7000 fluorescence spectrophotometer (Shimadzu, Kyoto, Japan) with the excitation and scanning wavelength of 280 nm and 300–450 nm, respectively.

#### 2.5.4. Fourier Transform Infrared (FT-IR) and Far-UV Circular Dichroism (CD) Spectra

The infrared spectra of flaxseed protein fractions were assessed using a Fourier transform-infrared spectrum (FT-IR) (Vertex 70, Bruker, Germany). In brief, the freeze-dried power samples were fully mixed into a KBr pellet (1%, *w*/*w*), and recorded in the wavelength range 4000–400 cm^−1^. The far-UV CD spectra were obtained using a Chirascan-plus spectropolarimeter (Applied Photophysics Ltd., Surrey, UK). The far-UV CD spectra were performed in a quartz cuvette of 2 mm with a protein concentration of 0.1 mg/mL in 50 mM PBS (pH 6.8). The samples were scanned from 190 to 260 nm at 25 °C. The contents of α-helix, β-strand, β-turns, and random coil were estimated from the far-UV CD spectrum using the deconvolution program (CDNN, version 2.1. Applied Photophysics Ltd., Surrey, UK).

### 2.6. In Vitro Antioxidant Activities and Free Phenolic Profiles of Flaxseed Protein Fractions

Flaxseed protein fractions were extracted with 5 mL of methanol aqueous solution (80%, *v*/*v*) by vortex mixing for 10 min and subsequent ultrasonic bath for 20 min. following centrifugation at 5000× *g* for 20 min. The in vitro antioxidant activities of extracts from flaxseed protein fractions were performed by 2,2-Diphenyl-1-picrylhydrazyl (DPPH) and ferric reducing antioxidant power (FRAP) methods [24]. The values of DPPH and FRAP were expressed as mg ascorbic acid equivalents (AAE)/100 g protein samples, respectively. The total phenolic acids and flavonoids of extracts were determined using the Folin–Ciocalteu and aluminum nitrate assays. The results were expressed as mg gallic acid (GAE) and rutin equivalents (RE) per 100 g protein sample (dry basis), respectively [25]. The free phenolic acids of extracts were analyzed using an Agilent 1290 ultrahigh performance liquid chromatograph (Agilent, Santa Clara, CA, USA) coupled with a PDA detector and ACQUITY UPLC^®^ BEH Shield RP18 column (2.1 × 100 mm, 1.7 μm). The results were quantified using the individual external standard and expressed as mg/100 g protein sample [26].

### 2.7. Foaming Properties of Flaxseed Protein Fractions

#### 2.7.1. Foaming Capacity and Stability

In brief, 15 mL of flaxseed protein dispersion (1.0%, *w*/*v*) was accurately added into a graduated container and homogenized at 10,000 rpm for 2 min. The foaming capacity (*FC*, %), and foaming stability were calculated as the following equations
(1)$$FC\;\left(\%\right)=V_{0}/V_{L}\times 100$$
(2)$$FS\;\left(\%\right)=V_{0}/V_{1}\times 100$$
where, *V_L_* was the volume of non-shearing protein dispersion, 15 mL; *V*_0_ was the volume of foams immediately after shearing; *V*_1_ was the volume of foams 30 min after shearing.

#### 2.7.2. Morphological Structure of Foams

Approximately 20 μL of foams were deposited on a glass slide with a groove and coverslip. The foam shape and size were acquired using an optical microscope (Scope. AI, Carl Zeiss, Oberkochen, Germany) equipped with a CCD camera (Jenoptik C14. Jenoptik Laser GmbH, Jena, Germany).

#### 2.7.3. Air–Water Interfacial Activity and Microrheological Behavior of Flaxseed Protein Fractions

The interfacial pressure (π) of flaxseed protein fractions at air–water interface was determined with a tensiometer (K100, Kruss, Hamburg, Germany) using the Wilhelmy plate technique. In brief, the platinum plate was immersed in 40 mL of flaxseed protein dispersion (1.0%, *w*/*v*) to a depth of 2 mm. The time-dependent π values were continuously recorded for 3600 s at 25 °C.

The microrheological behavior of flaxseed protein fractions was determined by a microrheometer Rheolaser Master^TM^ (Formulation, Toulouse, France). A cylindrical glass tube containing the freshly prepared sample (20 mL) was placed in the microrheometer chamber and analyzed for 3 h at 25 °C. The results were expressed as the elasticity index (EI), macroscopic viscosity index (MVI), and fluidity index (FI) using the software Rheosoft Master 1.4. Then, the storage modulus (G′) and loss modulus (G″) of samples were obtained in the range of 1–100 Hz when the measurement time was up to 10 min [27].

### 2.8. Emulsifying Properties of Flaxseed Protein Fractions

#### 2.8.1. Preparation of Emulsions

The flaxseed protein fractions were dissolved into 50 mM PBS (pH 6.8) at the concentration of 1.0% (*w*/*v*) by continuous stirring for 2 h at 25 °C and were then stored overnight at 4 °C for a complete hydration. In order to avoid the potential influences of high-pressure homogenization on the emulsifying properties of flaxseed protein fractions, the coarse emulsions were prepared by mixing 10% (*w*/*w*) flaxseed oil into protein dispersion using a high-speed homogenizer (IKA, T25, Staufen, Germany) at 10,000 rpm for 2 min.

#### 2.8.2. Physicochemical Properties and Physical Stability of Emulsions

The mean particle size, volume fraction, and zeta potential of emulsions were evaluated using a ZetaSizer Nano-ZS 90 (Malvern Instruments Ltd., Worcestershire, UK). The morphology of emulsion droplets was observed using optical microscope. After homogenization for 0 min and 10 min, 50 μL of the emulsions were immediately taken from the bottom of the beaker and diluted as 1:100 with 0.1% SDS solution. The absorbance of diluted emulsions was recorded at 500 nm. The emulsifying activity index (*EAI*) and emulsifying stability index (*ESI*) were calculated as the following equations [28]:(3)$$EAI\left(m^{2}/g\right)=\frac{2\times 2.303\times A_{0}\times DF}{C\times \varphi \times \theta \times 10,000}$$
(4)$$ESI\left(\min\right)=\frac{A_{0}}{A_{0}-A_{10}}\times 10$$
where *DF* was the dilution factor (100), *C* was the protein concentration (g/mL), *φ* was the optical path (1 cm) and *θ* was the oil volume fraction (0.25), *A*_0_ and *A*_10_ were the absorbance of the emulsions at 0 min and 10 min, respectively.

Then, the emulsions were subjected to the multiple light scattering (MLS) measurement (Turbiscan LAB, Formulaction, Toulouse, France). In brief, emulsions were transferred in a cylindrical glass cell and the curves of transmitted and backscattered light intensity versus the scanning height over a whole length of 40 mm were scanned at 25 °C for 30 min. The delta backscattering (ΔBS) was developed to evaluate determining emulsion stability by evolution of backscattering with time. The turbiscan stability index (TSI) was calculated from the BS of near-infrared light as a function of height.

#### 2.8.3. Microrheological Behavior of Emulsions and Oil–Water Interfacial Activity of Flaxseed Protein Fractions

The microrheological behavior of emulsions was determined by a microrheometer as described above. Then, the interfacial pressure (π) values of flaxseed protein fractions at oil–water interface were determined with a tensiometer using the Wilhelmy plate technique. First, the platinum plate was immersed in 14 mL of protein dispersion (1.0%, *w*/*v*) to a depth of 2 mm. Then, 40 g of flaxseed oil was added into the dispersion to create the oil–water interface. The time-dependent π values of flaxseed protein fractions at the oil–water interface were continuously recorded for 3600 s at 25 °C.

#### 2.8.4. Determination of Percentage of Adsorbed Proteins (AP%)

Percentage of adsorbed proteins were determined using the method described by Liang and Tang with slight modifications [22]. First, the 1mL emulsion was centrifuged at 8000× *g* for 20 min. Next, subnatant was carefully collected using a syringe, and protein concentration of the subnatant (*C_S_*) was determined with the BCA method using BSA as the standard. The adsorbed proteins (*AP*) were calculated as the following equation. Where *C*_0_ was protein concentration in the initial protein solutions.
(5)$$AP\left(\%\right)=\frac{C_{0}-Cs}{C_{0}}\times 100$$

#### 2.8.5. Microstructure of Emulsions

In brief, 2.0 μL of emulsions were frozen in liquid nitrogen, transferred into a chamber, cut into the cross section, and then sublimated at −80 °C and 1.3 × 10^−6^ mbar for 8 min using the PP3010T Cyro-SEM Preparation System (Quorum, Ringmer, UK). The in situ microstructure of lipid droplets and bulk continuous phase in emulsions were observed at 3 kV with magnification 2000×.

### 2.9. Statistical Analysis

The data were presented as mean ± standard deviations (*n* = 3) and carried out with SPSS 24 for Windows (SPSS Inc., Chicago, IL, USA). One-Way ANOVA, followed by Duncan test, was performed to analyze the significant differences between data (*p* < 0.05).

## 3. Results and Discussion

### 3.1. The Proximate Composition and Physicochemical Properties of Flaxseed Protein Fractions

As shown in Table 1, the highest moisture content was observed for FA, whereas the lowest value was found in FG, followed by FPI when subjected to the same freeze-drying condition. High and comparable crude protein contents were obtained for FPI, FG, and FA (>90%), which largely exceeded the results of protein fractions extracted from whole flaxseed [13]. In the current study, the excellent purity of protein fractions could be explained by the absolute removal of gum polysaccharides when dehulled flaxseed was selected as an extraction substrate [29]. Moreover, higher branched-chain, hydrophobic, and sulfur-containing amino acids, but lower negatively charged amino acids, were observed for FG in comparison to that of FA, which was in line with the findings reported by Madhusudhan and Singh [30]. Notably, the extremely low levels of total sugars were examined for FPI, FG, and FA due to the absolute removal of gum polysaccharides. FG had a higher content of total lipids when compared with that of FA (+56.00%, *p* < 0.05), followed by FPI (+42.67%, *p* < 0.05). The stronger surface hydrophobicity of FG might lead to easier oil–absorption capacity when compared with that of FA. The relatively low ash contents were determined for FA, FG, and FPI due to the dialysis process.

As seen in Figure 1a, a significantly larger hydrodynamic size was found for FPI, followed by FG when compared with that of FA (+4.21-fold, +1.18-fold; *p* < 0.05), which might be owing to the formation of protein aggregates between FA and FG [10]. The highest zeta potential value was observed for FPI when compared to those of FG and FA (+22.05%, +68.66%; *p* < 0.05). This could be explained by the mild conformation unfolding and deformation based on the intermolecular interactions between FA and FG. As presented in Figure 1b, an extremely lower protein solubility was found for FPI (−2.23-fold; *p* < 0.05), followed by FG (−0.91-fold; *p* < 0.05) when compared with those of FA (1.0%, *w*/*v*). FG displayed a highly stronger surface hydrophobicity than that of FPI and FA (+24.63%, +154.28%; *p* < 0.05). The basic subunits of FG with higher hydrophobic acids were usually buried in the inside of protein molecules in bulk aqueous phase. Thus, the surface hydrophobicity of FG depended on the outer acidic subunits oriented towards the aqueous phase due to the multi subunit tangling structure. The superior solubility of FA could be attributed to the single protein subunit with low molecular weight and hydrophobic amino acids. Upon the alkaline dissolution and acid precipitation process, the internalization of hydrophobic group might occur based on the noncovalent interactions between FG and FA, thereby resulting in mild reduction of the hydrophobic ability for FPI [31]. Concurrently, the formation of aggregates between FA and FG fractions definitely weakened the solubility of FPI, as evidenced by the largest hydrodynamic dimensions.

### 3.2. The Morphological and Structural Properties of Flaxseed Protein Fractions

As demonstrated in Figure 1c, FG displayed an approximate spherical morphology with relatively lager size distribution, further confirming the different degrees of aggregation between protein molecules. Comparatively, FA exhibited a large lamellar structure, which could be explained by the extended molecular conformation. Notably, FPI contained a relatively small lamellar strip structure (albumin), which was packed by the blurring spherical particles (globulin). Accompanied by the conformational extension and reconfiguration, the somewhat noncovalent interactions and complex aggregates between FA and FG might preferentially exist in FPI during the alkaline dissolution and acid precipitation. Undoubtedly, the surface morphology of FPI was partially inconsistent with the findings, as previously described in our study [23]. The gum polysaccharides with irregular block shape could naturally coexist in FA fraction during the extraction process, and thus affect the surface morphology of FPI obtained from whole flaxseed meal [7]. Moreover, the inconsistent isoelectric points between FA and FG also contributed to the different composition profiles in FPI obtained by the alkali dissolution and acid precipitation [32,33]. In addition, FPI could undergo varying structural alterations during different pretreatment and extraction process, thereby leading to different morphological characteristics compared with FA and FG prepared by ultracentrifugation and salting out, respectively [12,14].

As depicted in Figure 1d, FA had a major band at 10 kDa, followed by 18 and 20 kDa. In addition, an extremely weak intensity of band was also identified approximately at 35 kDa for FA, which might be explained by the coextraction of small amounts of FG due to the noncovalent interaction between them. By contrast, FG contained high intensity of bands at 35 and 18 kDa, followed by the light bands at about 55 and 20 kDa, respectively. However, FPI possessed similar subunit profiles but different subunit proportion when compared with that of FG, particularly for the evidently diminished intensity of band at 55 kDa and synchronously enhanced intensity of band at 35 kDa. Our findings were partly incompatible to the observation reported by Nwachukwu and Aluko, who prepared FA and FG from flaxseed protein meal, which could be further affected by the extraction matrix and methods [13]. Differential protein subunit profiles of albumin and globulin were obtained from other plant proteins, such as buckwheat and soapnut seeds [34,35].

The conformational properties of flaxseed protein fractions were sequentially analyzed by intrinsic fluorescence spectrum. As showed in Figure 1e, FPI presented the maximum fluorescence emission spectrum (λmax) at 331.6 nm. In comparison, FG exhibited slightly quenching maximum fluorescence intensity (FImax) at the similar λmax. However, the maximal value of FImax and 6.2 nm of red-shift of λmax were concurrently detected for FA in contrast to that of FG. Indeed, except for the high level of Trp, FA manifested flexible structural properties. Relatively, the closer chain packing of Trp residues within hydrophobic pocket and subsequent folded spatial conformation could be manifested for FG [36]. Although the gum polysaccharides were removed from protein fractions, the naturally coexisting phenolic compounds with varying abundances and profiles undoubtedly affected the spatial conformation of FA, FG, and FPI due to the noncovalent interaction between them [15].

The FT-IR spectrum of proteins possessed two primary features, the amide I (1620–1700 cm^−1^) and amide II (1500–1560 cm^−1^) bands raised from the specific stretching and bending vibrations of protein backbone C=O, C-N, and N-H. The amide I band of FPI had a blue shift of 1.9 cm^−1^ and 3.8 cm^−1^ relative to FG and FA. These indicated that the intermolecular hydrogen bonds were formed between the two components, and the electron cloud density of C=O decreased, which caused the absorption peak to shift to the lower wave number direction [37]. The detailed secondary structure content of different protein components was further obtained from the far-UV CD spectrum, as illustrated in Figure 1g. FPI contained 13.21% of α-helix, 40.44% of β-sheet, 15.58% of β-turns, and 30.75% of random coil, respectively, which was similar to that of FG. By contrast, obviously higher α-helix but lower β-sheet content was determined for FA when compared those of FG (*p* < 0.05), which could be explained by the more disulfide linkages and ordered structure as reported by Madhusudhan and Singh [30].

### 3.3. The Antioxidant Activities and Free Phenolic Acids of Flaxseed Protein Fractions

As depicted in Table 2, FA possessed the largest DPPH value, which was 5.49-fold higher than that of FG (*p* < 0.05), followed by FPI (−1.38-fold, *p* < 0.05). Similar changing trends were detected for the FRAP values of FA, FPI, and FG, revealing the specific accumulation of phenolic compounds during extraction process. As assessed by colorimetric methods, the contents of total phenolic acids and flavonoids in FPI reached 89.65 and 1.07 mg/100 g, respectively, which was obviously lower than the values of flaxseed protein prepared from whole flaxseed meals [23]. Indeed, flaxseed hulls also contained relatively abundant lignan oligomers in the secondary wall of sclerite cells, which consisted of secoisolariciresinol diglucoside (SDG), p-coumaric acid, ferulic acid, and herbacetin diglucosides [38]. The glycosidation and complex ester bonds in lignan oligomers could partially interfere with the contact between phenolic hydroxyl groups and free radicals. The steric hindrance effect of lignan could only produce relatively poor free radical scavenging activities when marginally migrated into flaxseed protein during extraction process [39]. Notably, the preferable accumulation of total phenolic acids and flavonoids was observed for FA in comparison to that of FG (+2.24-fold, +60.00%; *p* < 0.05). Nevertheless, equal contents of phenolic compounds were detected for both globulin and albumin prepared from flaxseed protein precipitated at the isoelectric point of 4.2 [31]. In fact, the different abundance and profiles of phenolic compounds in varying extraction substrates, such as whole/dehulled flaxseed, defatted flaxseed meals, flaxseed protein concentrates, etc., greatly affected the subsequent release and retention into FG and FA fractions. Thus, no signs of SDG, *p*-CouAG, and FeAG in lignans were identified for FPI, FA, and FG when the dehulled flaxseed was selected for extraction substrate, thereby resulting in no contribution to the antioxidant activities of protein fractions.

Instead, several free phenolic acids, including gallic acid, coumarin, protocatechuic acid, vanillin, sinapic acid, sallcylic acid, caffeic acid, cinnamic acid, syringic acid, and vanillic acid were identified in FA, accounting for 86.82%, 3.52%, 3.27%, 2.87%, 1.33%, 0.70%, 0.56%, 0.38%, 0.36%, and 0.19% of total amounts (69.06 mg/100g), respectively. By comparison, largely lower contents of free phenolic acids were obtained for FG when compared with that of FA (−72.84%, *p* < 0.05). In particular, higher proportion of gallic acid, protocatechuic acid, sinapic acid, sallcylic acid, syringic acid, vanillic acid, and coumarin (86.94%, 4.42%, 1.92%, 1.65%, 1.44%, 1.23%, 1.17%), but lower proportion of vanillic and cinnamic (0.80%, 0.43%) were identified in FG. However, the free caffeic acid was not detected in FG, suggesting obviously preferential migration into FA during extraction process. As previously reported, the free phenolic acids mainly coexisted with the extract supernatant of flaxseed protein and possessed higher antioxidant activity than bound phenolic acids detained in residue remaining after extraction [40]. Indeed, there was still no definite information about the specific in situ location of free phenolic acids in dehulled flaxseed [41]. Moreover, the dislocation migration of free phenolic acids between oleosins and storage proteins might occur owing to the newly established contact and strong noncovalent interaction between them upon high-speed shearing of flaxseed [40]. The lower molecular weight, more stretching conformation and superior molecular polarity contribute to the favorable migration of free phenolic acids into FA, inevitably leading to a weaker retention in FG following sequential extraction [34,42]. Most importantly, the proportion and noncovalent interactions between FA and FG definitely complicated the retention properties of free phenolic acids into FPI during the extraction process.

### 3.4. The Foaming Properties of Flaxseed Protein Fractions

The foam height can visually reflect the volume of foam produced by mechanical shearing of a bulk liquids and was considered as an index of foaming capacity of proteins. As shown in Figure 2a,b, the maximum foam volume was identified for FA, whereas the minimum value of foam was observed for FG (1.0%, *w*/*v*). Thus, the presence of FA fraction largely determined the foaming potential of FPI, which was undoubtedly inferior to that of individual FA (−23.9%, *p* < 0.05). The flexible spatial conformation, superior solubility, and lower molecular weights could enhance the migration efficiency of FA at the air–water interface, leading to the preferable foaming ability [43]. In comparison, the poor foaming activity of FG might be explained by the slow interface migration from the bulk aqueous phase into air/water interface due to the higher hydrophobic amino acids and molecular weights, respectively. As for FPI, the desirable interface absorption behavior of FG was positively driven by the coexisting low amounts of FA, which might occur due to the noncovalent interaction between them. Then, the foaming stability was evaluated by time-dependent subduction of foam height and corresponding changes in foam morphology (Figure 2b,c). After standing for 30 min at 25 °C, the foam volume produced by FPI, FG, and FA decreased by 22.3%, 57.5%, and 42.2% (*p* < 0.05), respectively, suggesting a desirable foaming stability of FPI relative to FA and FG. As evidenced by the microscopic imaging (Figure 2e), the foam freshly prepared by FG was distributed over a wide range of size, which inevitably could lead to an immediate coalescence or collapse after formation. In contrast, the sizes of foams freshly prepared by FA were distributed in a relatively narrow range, followed by FPI. Then, the foams obtained by FA were mainly subjected to rapid drainage, but no coalescence or collapse within 30 min of standing. However, a relatively slower drainage, coalescence, and collapse was displayed for foams prepared from FPI. Undoubtedly, the foaming stability of FPI was largely attributed to the coexistence of FA and FG with appropriate proportion at air/water interface. In depth, the interface intermolecular interactions between FA and FG in FPI could partially narrow the interface “holes”, and subsequently suppress the gas migration between foams, leading to relative stability of foams prepared by FPI.

The abilities of flaxseed protein fractions to be absorbed at the air/water interface were further evaluated through time-dependent alteration of surface tension (π) values. As presented in Figure 2f, the initial π value of FPI (1.0%, *w*/*v*) was 41.60 mN m^−1^, which slightly decreased with the equilibrium time extended to 3600 s. By comparison, FG possessed comparable initial π value, but displayed relatively stronger decline with the increase in equilibrium time (−6.39%, *p* < 0.05). However, FA manifested the lowest initial π value, but was comparable to that of FG when the equilibrium time reached 3600 s. The excellent performance of FA for early migrating into the air/water interface undoubtedly explained its desirable foaming capacity. However, the slight reduction of interfacial tension over time might suggest the relatively limited conformation reorganization at air/water interface due to the stretched conformation and weak intermolecular interactions, thereby leading to fast drainage upon standing [44]. In comparison, FG was unable to achieve early adsorption at air/water interface due to the multi subunit crosslinking structure, high molecular weight, and hydrophobicity. Instead, effective conformation adaptability and strong intermolecular crosslinks might inevitably occur for FG, leading to the formation of densely packed viscoelastic films. The higher initial value of interfacial pressure and slight reduction over time for FPI suggested that the low amounts of FA and noncovalent interactions with FG sufficiently resulted in low interfacial diffusion, adsorption, and conformation reorganization [43].

The EI calculated from the inverse of the mean square displacement (MSD) for a decorrelation period can quantify the elasticity of samples. The MVI, as the inverse slope of the MSD curves in double liner coordinates, can characterize the viscosity of samples. The fluidity index (FI) curve can reflect the fluidity properties of samples. The greater the fluidity of a sample, the weaker its viscosity. As seen in Figure 3a–c, FPI and FA possessed obviously higher viscoelastic properties, which might contribute to the delaying fluid drainage between bubbles with the interface shrunk, thereby accounting for their more desirable foaming ability when compared with that of FG. FPI displayed comparable fluidity to FA and FG, and then steeply dropped during the late test period, which might result from the intermolecular interactions between FA and FG in aqueous phase under no shearing condition, and partially explain the relatively ascendant foaming stability dependent on the densely packed monolayers. Actually, the naturally occurring phenolic compounds might also affect the structural adaptability and subsequent gas-holding capacity of protein fractions at the air–water interface. In particular, the coexistence of free phenolic acids in FA might compensate for its molecular flexibility and tend to form rigid and compact interface with desirable stability [45]. It can be seen that for all protein solution (1%), storage modulus (G′) was beyond the loss modulus (G″) in the range of 1–10 Hz, then G″ became greater than G′ with a continuous increase in frequency (Figure 3d). The bulk phase response was dominated by viscous effects, indicting that the protein in the bulk phase was adsorbed or aggregated to the interface so that loss modulus increased faster than the storage modulus [46,47].

### 3.5. The Emulsifying Properties of Flaxseed Protein Fractions

#### 3.5.1. The Physicochemical Properties and Physical Stability of Emulsions

As illustrated in Figure 4a,b, the mean lipid droplet sizes and zeta potential values of emulsions constructed by FPI were 1445.33 nm and −11.60 mV, respectively. By comparison, the mean particle size of emulsions fabricated by FG increased by 20.92% (*p* < 0.05), which was accompanied by the decrease in zeta potential value (−19.40%, *p* < 0.05). Relatively, FA behaved with superior emulsifying capacity as evidenced by the minimum particle dimensions, but comparable charge density to that of FPI. These indicated that the molecular conformation of FG in FPI might be interfered due to the noncovalent interactions with FA, which was favorable for the emulsion stability via the electrostatic repulsion between adjacent lipid droplets. According to the results from microscopic examination, FA produced the emulsion with relatively small and evenly distributed lipid droplets, whereas the lipid droplets produced by FG manifested larger mean particle sizes. As expected, the lipid droplets with more heterogeneous sizes had been observed for emulsion droplets stabilized by FPI. In particular, the beaded small lipid droplets and subsequent close contacts with large lipid droplets might weaken the emulsifying stability of FPI [29]. The different molecular composition and conformation of FA and FG definitely complicated the interfacial behavior and rheological properties of emulsions, which largely determined the physical stability of emulsions formulated by flaxseed protein fractions. 

As depicted in Figure 4c, FG and FPI had comparable but significantly lower EAI values when compared with that of FA (−45.61%, −39.63%, *p* < 0.05). However, the maximum ESI value was observed for FG, followed by FPI, whereas FA displayed the lowest levels of ESI. These indicated that the emulsifying activity and stability of FPI were contributed by the FA and FG, respectively. Consistent with the above results, the emulsion stability results obtained by the TSI values were as follows: FG > FPI ≈ FA (Figure 4d). As shown in Figure 4e, the ΔBS lines of emulsions formulated by FPI dramatically decreased with extending time in the measurement cell height range of 1.5 mm to 32 mm. Simultaneously, different increments in ΔBS curves of emulsions were observed in the measurement cell range of 32~38.5 mm, which was accompanied by a slight decline in ΔBS curves in the topmost part of the measurement cell (38.5~40 mm). These further confirmed that the instability of emulsions prepared by FPI was primarily due to the serious flocculation and coalescence of lipid droplets, followed by the lipid droplet flotation. As to the emulsion prepared by FG, the relatively smaller decrease in the middle curve of ΔBS revealed a weaker lipid droplet flocculation, coalescence, and floatation. In comparison, the large-scale reduction of ΔBS lines in the bottom part of measurement cell (0~10 mm) and synchronous promotion in the top part of measurement cell (32~37 mm) were found for the emulsions constructed by FA, manifesting the maximum lipid droplet flotation and oil perspiring, followed by flocculation and coalescence [48]. Similarly, Nwachukwu and Aluko found that the FG had better emulsifying stability than FA at neutral environment and specific concentration [13]. As depicted in Figure 4f, the relatively fast phase separation of emulsions prepared by FPI, FA, and FG indicated that the introduction of emulsifier, such as the gum polysaccharides and phospholipids, was still necessary to produce the gel network structure in bulk phase, applied for direct interface coating or narrowing the pore sizes of lipid droplets in emulsions obtained from FPI, FA, and FG [49].

#### 3.5.2. The Interface Activities of Flaxseed Protein Fractions

The time-dependent tracking of interfacial tension for protein fractions mainly reflected their differences in instantaneous migration and adsorption capacity towards the oil–water interfaces. As presented in Figure 4f, the initial surface tension (π) value of FPI dispersion (1.0%, *w*/*v*) reached 16.69 mN m^−1^, which gradually descended and achieved an ultimate reduction by 15.94% (*p* < 0.05) with the equilibrium time extended to 3600 s. By contrast, a lower initial π value (15.45 mN m^−1^) with an almost similar ability to reduce the interfacial tension over time was observed for the FA dispersion (−15.48%, *p* < 0.05). However, FG dispersion exhibited the lowest initial π value (14.95 mN m^−1^), and the maximum interfacial dropping trend (−17.73%, *p* < 0.05). It was considered that the primary driving force for the migration and absorption at the oil–water interfaces was the solubilization of nonpolar regions for protein molecules in oil phase. The higher affinity of FG to the oil–water interface could be demonstrated by the stronger hydrophobicity than that of FA fraction. As shown in Figure 4g, the percentages of adsorbed protein (AP%) at the interface of lipid droplets were approximately 43.44%, 36.58%, and 12.83% of the total protein for FPI, FG, and FA dispersion (1.0%, *w*/*v*), respectively. Thus, the more substantial conformation reorganization and greater adsorption capacity could also be elucidated from the effective interface adsorption kinetics and molecule loads for FG when compared with that of FA [50]. Importantly, the highest interface adsorption but limited interface deformation for FPI might be attributed to the formation of specific interface membrane due to the intermolecular interaction between FG with FA upon high-speed shearing.

#### 3.5.3. The Microrheological Properties of Emulsions

According to Figure 5a,b, the emulsions stabilized by FPI and FG (1.0%, *w*/*v*) displayed the relatively low and almost equivalent viscoelasticity during the test period (0~3 h). By contrast, the emulsion produced by FA displayed the relatively higher viscoelasticity, consistent with the trend of EI and MVI curves of the emulsions produced by FPI and FG. As presented in Figure 5c, the results of FI were consistent with the MVI for emulsion prepared by FA, FG, and FPI. As illustrated in Figure 5d, the values of loss modulus (G″) of emulsion constructed by FPI apparently exceeded the storage modules (G′) over the frequency from 8.6 to 100 Hz, exhibiting the relatively weak gelling properties. Undoubtedly, the viscous properties of emulsion stabilized by FPI were largely contributed by the FA, but not FG fraction according to the G″ and G′ values of emulsion produced by FG and FA. Thus, it could be considered that the higher proportion of FA in bulk aqueous phase primarily determined the rheological behavior of emulsions.

#### 3.5.4. The Microstructure of Emulsions

We further explored the microstructure of lipid droplets and bulk continuous phase in emulsions prepared by FA, FG, and FPI. As observed in Figure 6, a flimsy but relatively compact interfacial film was observed for the lipid droplets stabilized by FPI ranging from 4 μm to 20 μm based on the cryo-SEM imaging. Notably, FPI produced no spatial network structure in bulk continuous phase, which might be due to its desirable emulsifying capacity. Undoubtedly, the multiple close contacts between lipid droplets produced by FPI definitely contribute to the emulsion instability via the flocculation and coalescence, which further supported the results from the Turbiscan analysis. The small content of albumin in rapeseed protein isolate was demonstrated to play a major role in the formation of interface structure [50]. On the contrary, FG produced the lipid droplets with heterogeneous dimensions (2~20 μm), which was attached on the compact single, double, or multilayer lamellar gel network structure in bulk aqueous phase dependent on the particle sizes. These indicated that FG possessed the excellent potential of restricting the movement of lipid droplets via the interlayer anchoring but not direct interface coating. Notably, both the interface film of lipid droplets and lamellar gel structure of bulk aqueous phase were emerged for emulsion prepared by FA. In particular, the quite loose and porous interface microstructure of lipid droplets could be explained by the low interfacial adsorption and weak intermolecular interactions, as evidenced by the information from the tensiometer analysis. The relatively dispersed lipid droplets in bulk aqueous phase might explain the fast oil perspiring but relatively slow flocculation and coalescence rates when subjected to standing. The in situ interfacial microstructure of lipid droplets and continuous phase intuitively explained the different emulsifying properties of FG and FA in FPI, contributing to the specific stabilizing and destabilization behavior of emulsions.

## 4. Conclusions

In conclusion, FA possessed smaller particle size and desirable protein solubility due to lower molecular weight and surface hydrophobicity when compared to FG. Moreover, FA and FG manifested large lamellar and nearly spherical structure, respectively, which led to partial structure alteration for FPI due to the noncovalent interactions between them. The relatively flexible spatial conformation of FA resulted in favorable retention of free phenolic acids and increase in antioxidant activities compared to those of FG. Moreover, the foaming properties of FA were obviously superior to those of FG due to the effective adsorption behavior at air–water interface and higher viscoelastic properties of bulk phase. Notably, FA produced the emulsion droplets via direct interface coating, which was easily destabilized due to a quite loose and porous interface. FG exerted the interlayer anchoring and restricted the movement of lipid droplets by the spatial network structure. Notably, the emulsifying capacity and stability of FPI were determined by the synergistic interface and rheological behavior of FA and FG. Thus, the tailed retention of albumin could serve as an effective strategy to improve the selected techno-functionality of flaxseed protein based on the specific interaction with globulin fraction and naturally occurring phenolic acids.

## Figures and Tables

**Figure 1 foods-11-01820-f001:**
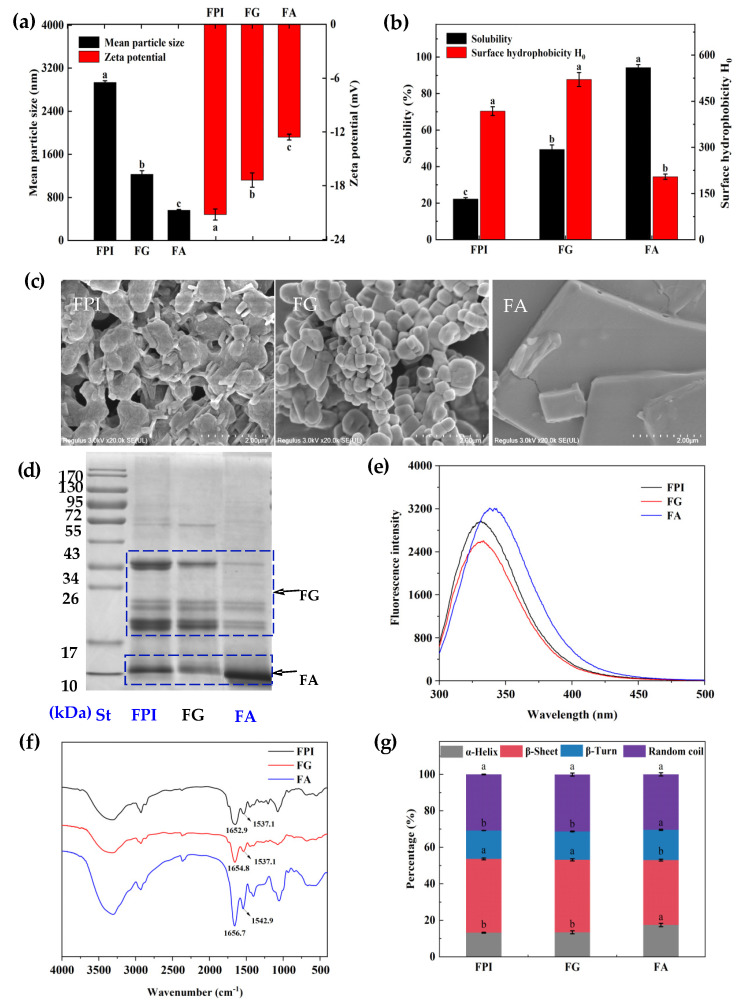
The physicochemical, morphological, and structural properties of FPI, FA, and FG. (**a**): the mean particle size (black bar) and zeta potential (red bar); (**b**): the solubility (black bar) and surface hydrophobicity (red bar); (**c**): the SEM imaging; (**d**): the SDS-PAGE analysis; (**e**,**f**): the intrinsic fluorescence and FTIR spectra; (**g**): the secondary structure obtained from CD spectra. Different alphabets in same index indicated significant differences at the *p* < 0.05 level. FPI: flaxseed protein isolate; FG: flaxseed globulin; FA: flaxseed albumin.

**Figure 2 foods-11-01820-f002:**
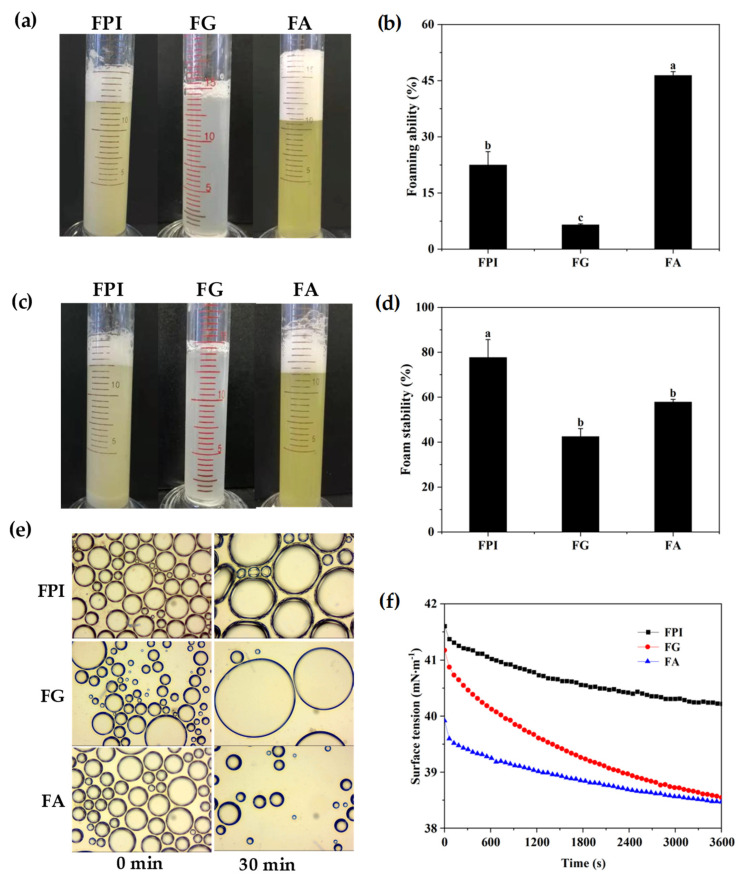
The foaming properties of FPI, FG, and FA. (**a**,**c**): photograph of foam formed by FPI, FG, and FA at 0 min and 30 min; (**b**): the foaming ability; (**d**): the foam stability; (**e**): the morphology of foams immediately after shearing and standing for 30 min; (**f**): the interfacial tension at air–water interface. Different alphabets in same index indicated significant differences at the *p* < 0.05 level. FPI: flaxseed protein isolate; FG: flaxseed globulin; FA: flaxseed albumin.

**Figure 3 foods-11-01820-f003:**
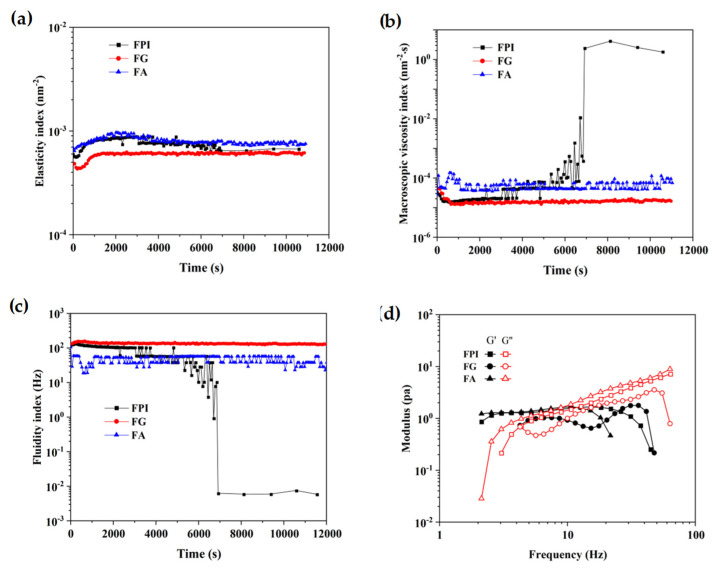
The microrheological properties of FPI, FG, and FA. (**a**): the elasticity index; (**b**): the macroscopic viscosity index; (**c**): the fluidity index; (**d**): storage modulus (G′) and loss modulus (G″). FPI: flaxseed protein isolate; FG: flaxseed globulin; FA: flaxseed albumin.

**Figure 4 foods-11-01820-f004:**
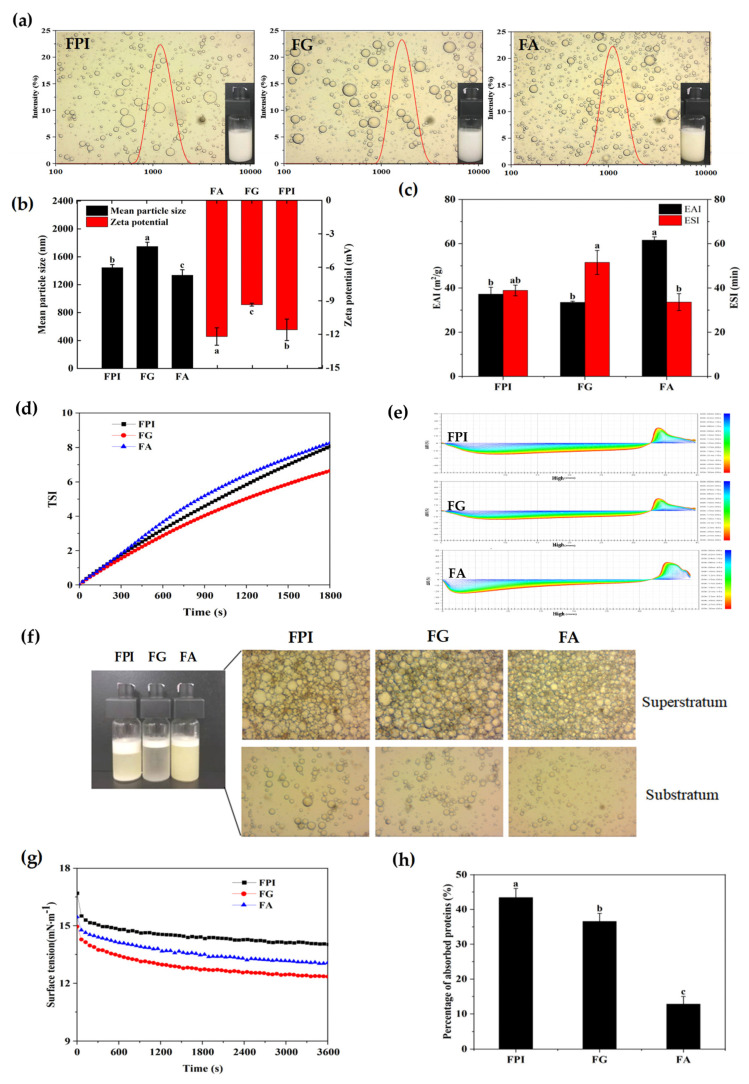
The physicochemical stability of emulsions constructed by FPI, FG, and FA. (**a**): the visual observation, optical micrographs, and particle size distribution; (**b**): the mean particle size (black bar) and zeta potential (red bar); (**c**): the EAI (black bar) and ESI (red bar); (**d**): the Turbiscan stability index (TSI); (**e**): the delta backscattering (ΔBS) curves; (**f**): the optical micrographs (stored at 4 °C for 48 h); (**g**): the surface tension at the oil–water interface; (**h**): the percentage of adsorbed protein fractions. Different alphabets in the same index indicated significant differences at the *p* < 0.05 level. FPI: flaxseed protein isolate; FG: flaxseed globulin; FA: flaxseed albumin.

**Figure 5 foods-11-01820-f005:**
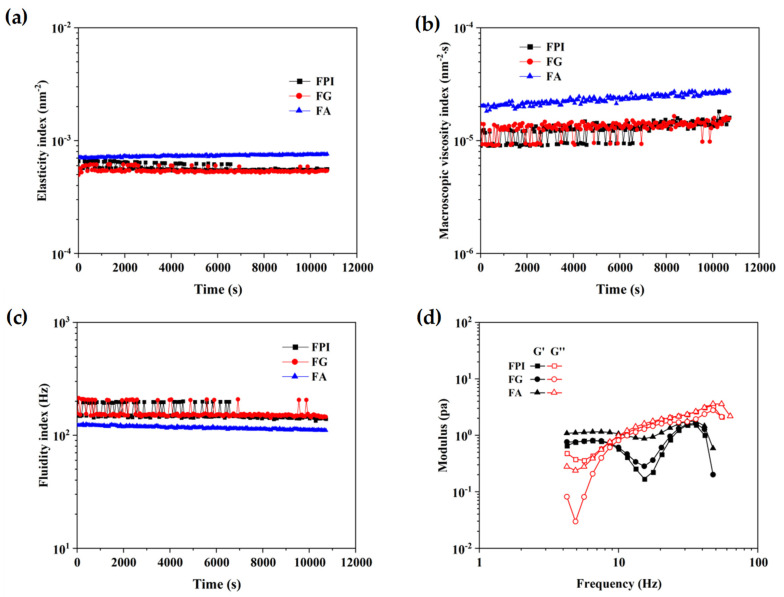
The microrheological properties of emulsions constructed by FPI, FG, and FA. (**a**): the elasticity index; (**b**): the macroscopic viscosity index; (**c**): the fluidity index; (**d**): the storage modulus G′ and loss modulus G″. FPI: flaxseed protein isolate; FG: flaxseed globulin; FA: flaxseed albumin.

**Figure 6 foods-11-01820-f006:**
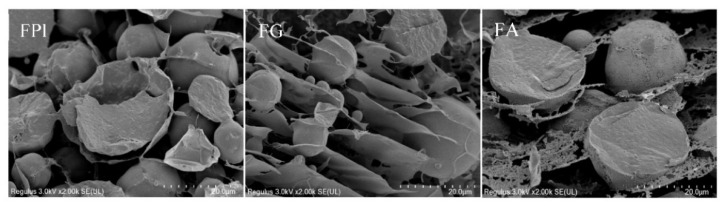
The cryo-SEM imaging of emulsions constructed by FPI, FG, and FA. FPI: flaxseed protein isolate; FG: flaxseed globulin; FA: flaxseed albumin.

**Table 1 foods-11-01820-t001:** The proximate composition of FPI, FG, and FA.

Parameters	FPI	FG	FA
Moisture (%)	3.43 ± 0.02 ^b^	2.09 ± 0.03 ^c^	4.61 ± 0.09 ^a^
Total sugars (%)	1.60 ± 0.54 ^b^	0.64 ± 0.20 ^c^	2.36 ± 0.33 ^a^
Crude proteins (%)	91.41 ± 1.41 ^a^	93.28 ± 1.09 ^a^	90.49 ± 1.25 ^a^
Total lipids (%)	2.14 ± 0.13 ^a^	2.34 ± 0.10 ^a^	1.50 ± 0.05 ^b^
Ash (%)	1.43 ± 0.09 ^b^	1.65 ± 0.17 ^a^	1.04 ± 0.13 ^c^
Amino acid composition (%)			
Aspartic acid	10.73	9.80	9.32
Threonine	3.68	3.40	3.13
Serine	5.07	4.71	4.59
Proline	4.46	2.49	4.27
Glutamic acid	17.80	19.57	24.14
Glycine	5.06	5.73	6.24
Alanine	5.30	5.56	4.66
Cysteine	0.25	1.52	0.33
Valine	6.73	8.11	5.65
Methionine	2.25	2.45	1.95
Isoleucine	5.56	6.41	4.66
Leucine	6.55	7.33	6.49
Tyrosine	3.80	2.95	3.51
Phenylalanine	6.29	4.43	4.89
Hlstidine	2.49	2.21	1.80
Lysine	3.64	3.74	4.78
Argnine	10.34	9.59	9.58
Tryptophan	-	-	-
BCAA	18.84	21.85	16.8
HAA	26.64	31.38	23.74
NCAA	28.53	29.37	33.46
PCAA	16.47	15.54	16.16
SCAA	2.5	3.97	2.28

Means with different letters (a, b, c) on the same line were significant differences at *p* < 0.05 level. FPI: flaxseed protein isolate; FA: flaxseed albumin; FG: flaxseed globulin; BCAA: branched-chain amino acids; HAA: hydrophobic amino acids; NCAA: negatively charged amino acids; PCAA: positively charged amino acids; SCAA: sulfur-containing amino acids; -: not detected.

**Table 2 foods-11-01820-t002:** The phenolic acid profiles and antioxidant activities of FPI, FG, and FA.

		FPA	FG	FA
Total phenolic acids (mg/100 g)		89.65 ± 9.65 ^b^	81.83 ± 1.99 ^b^	264.85 ± 13.22 ^a^
Total flavonoids (mg/100 g)		1.07 ± 0.02 ^a^	0.30 ± 0.04 ^b^	0.48 ± 0.01 ^b^
DPPH (mg AAE/100 g)		51.52 ± 3.23 ^b^	22.30 ± 0.44 ^c^	122.45 ± 4.41 ^a^
FRAP (mg AAE/100 g)		56.67 ± 3.55 ^b^	17.16 ± 0.62 ^c^	101.19 ± 4.59 ^a^
Free phenolic acids (mg/100 g)	Gallic acid	19.13 ± 0.25 ^b^	16.31 ± 0.31 ^c^	59.96 ± 0.27 ^a^
	Protocatechuic acid	2.70 ± 0.01 ^a^	0.83 ± 0.01 ^b^	2.26 ± 0.03 ^a^
	Vanillic acid	0.11 ± 0.01 ^c^	0.15 ± 0.01 ^a^	0.13 ± 0.01 ^b^
	Caffeic acid	0.26 ± 0.01 ^b^	-	0.39 ± 0.01 ^a^
	Syringic acid	0.05 ± 0.01 ^b^	0.27 ± 0.05 ^a^	0.25 ± 0.02 ^a^
	Vanillin	0.52 ± 0.03 ^b^	0.23 ± 0.02 ^b^	1.98 ± 0.02 ^a^
	Sinapic acid	0.29 ± 0.03 ^c^	0.36 ± 0.01 ^b^	0.92 ± 0.02 ^a^
	Coumarin	0.45 ± 0.01 ^b^	0.22 ± 0.01 ^b^	2.43 ± 0.03 ^a^
	Sallcylic acid	0.59 ± 0.02 ^a^	0.31 ± 0.01 ^c^	0.48 ± 0.01 ^b^
	Cinnamic acid	0.20 ± 0.01 ^a^	0.08 ± 0.01 ^b^	0.26 ± 0.01 ^a^

Means with different letters (a, b, c) on the same line were significant differences at *p* < 0.05 level. FPI: flaxseed protein isolate; FA: flaxseed albumin; FG: flaxseed globulin; FPP: flaxseed polyphenols; -: not detected.

## Data Availability

All data are provided in the manuscript.

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
