# Peer review of "Comparative Composition Structure and Selected Techno-Functional Elucidation of Flaxseed Protein Fractions"

_foods, 2022, doi:10.3390/foods11131820_

Round 1
Reviewer 1 Report
This study systematically investigated the physicochemical properties and functionalities of flaxseed proteins and employed extensive techniques in determining these characteristics. However, the methods employed are not well justified and the discussion lacks scientific accuracy.
- Figure 1d: why the thickest band in FA is not shown in the FPI sample? The FPI should contain both FG and FA. Also, please point out which bands represent which fractions in the figure.
- Please pay attention to the graph organization. For example, in figure 4 title, there is no indication for (e) and (f); in figures 1a and 1b, there is no indication regarding the meaning of bars with different colors
- 3.4. “thus, the mass ratio…” what is the mass ratio?
- 3.4. “The poor foaming properties of FG might be explainedby the limited solubility in bulk aqueous phase, which could easily…” if the low foaming capacity is caused by solubility issues, why the least soluble FPI produced better foam.
- 3.4. “Undoubtedly, the foaming capacity and stability of FPI were largely attributed to thecoexistence of FA and FG.” How come this is true when FG was just proved not to have ideal foaming properties
- 3.4 “As presented in Figure 2d, the initial π…”. I believe it is figure 2f you are referring to
- 3.4 “Instead, an effective conformational reorganization and strongmolecule crosslinks inevitably occurred for FG following the limited interfacial adsorpotion”. What’s the evidence and why not try to utilize the many tests conducted to analyze the results?
- figure 3. Please indicate what are the samples for this assay. I guess foams?
- “While the lower fluidity ofFPI may result from its insoluble protein.” why insoluble proteins would lower the fluidity of FPI? Did the protein precipitate?
- “FPI possessed a desirable emulsifying capacity but inferior emulsifying stability” please check if the use of “inferior” is correct, which appeared a few times in the MS.
- 3.5.1 firstparagraph: could you please explain why the emulsifying activities varied among different protein fractions? e.g., instead of stating their differences in zeta potential, explain how zeta potential is related to emulsification properties. The same for the second paragraph, please explain the mechanisms of emulsion destabilization and why some protein fractions produced emulsions with better stability while others do not?
Reviewer 2 Report
The manuscript present Flaxseed Protein Fractions functional properties. This manuscript need changes which are mention below
Abstract: Extensive revision is required in abstract, as the present sentences sounds noisy during reading. There are grammatical errors and need to minimize the sentences in length.
Keywords: Good enough
Introduction: Introduction is supported with nice arguments, some information need to address here
1. What is the % of protein in flaxseed?
2. Describe the arguments of individual protein fraction utilization, no such information has been provided.
3. No emulsion product or utilization of protein emulsion has been discuss. Please elaborate the point.
4. In last paragraph please add one or two sentences, why plant protein is important
5. Please add/indicate and compare your work with pervious published paper
Methods and materials:
The authors produced emulsion in 2.8, however
1. The authors did not mentioned emulsifying activity index (EAI) and stability index (ESI).
I suggest to add the above activities to compare the other parameters of emulsion and to understand the correlation.
2. What was the purity of protein fractions?
3. Is the emulsion stabilized by protein fractions only? Or added additives?
4. How the rheological parameters were measured? Nothing such has been reported in methods section.
5. What was the apparent viscosity of emulsion? Are the emulsion present Newtonian or non-Newtonian behavior? What was the flow index? Please add these parameters for emulsion.
Results and discussion: Discussion upon figures and table are fair and according to findings. Add more of information for droplet sizes and emulsion parameters.
1. The FTIR Spectra seems not fair enough, what is the difference between different bands?
2. Figure 5. Difficult to understand, why not compare data in tables?
3. No emulsion micro photos made after storing emulsion for 2 to 5 hours.
4. How the coalescence was stopped in emulsion?
5. Please add more of discussion on SEM, FTIR and solubility of protein fractions.
Conclusion: Conclusion fine
Round 2
Reviewer 1 Report
1. Figure 2 title: “the forming properties” or foaming properties? Plus, “a-d: the foaming ability and foam stability” is not detailed enough. The content of each figure should have been noted.
2. 3.4. the revised sentence: “Thus, the content of FA fraction largelydetermined the foaming potential of FPI, which was undoubtedly inferior to that of individual FA (-23.9%, p < 0.05).” please put a space between largely and determined. The effect of FA content was not studied in this research so it may be more appropriate to change this sentence to “the presence of FA…”.
3. 3.4. The added “…The desirable interface absorption behavior of FG driven by small amounts of FA in FPI might be occurred due to the noncovalent interaction between them, leading to the formation of with an insoluble protein particles” is irrelevant to my previous question because your statement focused on the improvement of FG interfacial behavior rather than explaining the relationship between solubility and foaming stability. It is not necessary to speculate “the formation of with an insoluble protein particles” if the authors do not think insolubility is the factor impairing foaming. The sentence has multiple grammatic errors, e.g. might be occurred, formation of with…. Plus, what is the evidence that adsorption rate of FG was enhanced? I would suggest deleting this particular sentence.
4. 3.4 “…foams, and just lead to the coarsening of foams prepared by FPI.” This sentence seems to be contradictory to the major point of this paragraph. “coarsening” indicates the foams is not as good but this paragraph aimed to establish that FPI has ideal foaming stability.
Reviewer 2 Report
Authors revised the manuscript as per suggested comments, Manuscript is clear for publication
Author Response
Thanks for your suggestion.
